# LANGUAGE MODEL DETECTORS ARE EASILY OPTIMIZED AGAINST

**Charlotte Nicks, Eric Mitchell, Rafael Rafailov, Archit Sharma,**
**Christopher D. Manning, Chelsea Finn, Stefano Ermon**
Stanford University
`cnicks13@stanford.edu`

## ABSTRACT

The fluency and general applicability of large language models (LLMs) has motivated significant interest in detecting whether a piece of text was written by a language model. While both academic and commercial detectors have been deployed in some settings, particularly education, other research has highlighted the fragility of these systems. In this paper, we demonstrate a data-efficient attack that fine-tunes language models to confuse existing detectors, leveraging recent developments in reinforcement learning of language models. We use the 'human-ness' score (often just a log probability) of various open-source and commercial detectors as a reward function for reinforcement learning, subject to a KL-divergence constraint that the resulting model does not differ significantly from the original. For a 7B parameter Llama-2 model, fine-tuning for under a day reduces the AUROC of the OpenAI RoBERTa-Large detector from 0.84 to 0.63, while perplexity on OpenWebText increases from 8.7 to only 9.0; with a larger perplexity budget, we can drive AUROC to 0.30 (worse than random). Similar to traditional adversarial attacks, we find that this increase in 'detector evasion' generalizes to other detectors not used during training. In light of our empirical results, we advise against continued reliance on LLM-generated text detectors. Models, datasets, and selected experiment code will be released at https://github.com/charlottttee/llm-detector-evasion.

## 1 INTRODUCTION

Large language models (LLMs) can produce high-quality text in a wide variety of settings (Brown et al., 2020; Bubeck et al., 2023). Access to such powerful LLMs has expanded rapidly; for anyone hoping to generate machine-written text, a plethora of free and low-cost options exist. The usage of such models has become endemic to classrooms, news outlets, social media platforms, and other domains. This rapid development has led to several objections to widespread use of LLMs, including moral qualms with data procurement, questions regarding the quality of machine-generated text, issues with LLMs outputting inaccurate information (hallucinations), and fear that the availability of LLMs may force people in a variety of roles to reevaluate their day-to-day practices.

These concerns have led to a significant amount of research and commercial product offerings for detecting machine-generated text (e.g. Gehrmann et al., 2019; Solaiman et al., 2019; Mitchell et al., 2023). While the proliferation of such detection techniques may plausibly quell such fears in the short term, can we continue to rely on this paradigm to detect machine-generated texts?

Existing detectors typically yield a scalar score. An adversary might therefore hope to fine-tune a language model to optimize this score, such that the model outputs are less detectable. By explicitly training to generate samples that confuse the classifier, the adversary could amortize the process of finding an adversarial attack: the fine-tuned LM would directly produce them. Text generation is a fundamentally discrete, non-differentiable domain, so we cannot optimize for such an attack directly; however, reinforcement learning (RL) can provide a framework for fine-tuning LLMs to optimize such blackbox scores.

This motivates several questions we study in this work: Can RL methods be used for making LLMs less detectable? How does detectability trade-off with other metrics like perplexity? How does

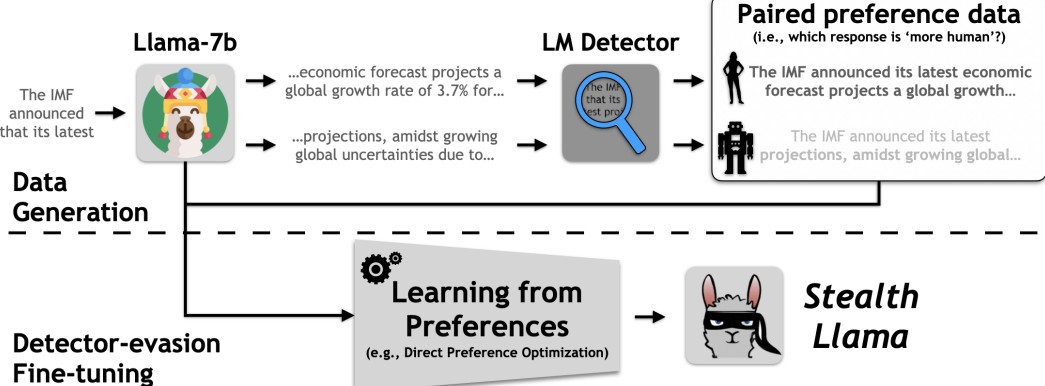

Figure 1: **We present results of a red-teaming effort that optimizes against machine-generated text detectors.** Top: We first generate a preference dataset that "prefers" generations that are more human according to the detector. Bottom: We fine-tune the model to optimize these preferences.

detectability scale with the size of preference datasets (or equivalently, with query budget to a detector), and is it feasible to train popular LLMs to be undetectable against commercial detectors on a limited budget? Does training against one detector reduce detectability under other detectors?

To study these questions, we consider a simple RL pipeline that uses direct preference optimization (DPO) (Rafailov et al., 2023). We use the score from the detector to construct a pairwise preference dataset where the completion with the higher 'humanness' score is marked as preferred, and the LLM is fine-tuned on this dataset using DPO. See Figure 1 for an overview of this data-generating and training pipeline. In contrast to prior work, such an approach does not require the use of human paraphrasers or a paraphrasing model (Krishna et al., 2023) and adds no overheard during inference.

The main contribution of this paper is a detailed empirical study on the ease of evading language model detectors by optimizing against them. Our experiments find that a simple DPO-based pipeline produces consistent reduction in detectability against various detectors. We can achieve AUROC metrics below 0.5 against several strong public and commercial detectors, indicating worse than random chance detector performance on the fine-tuned model and close to random chance performance on several additional detection algorithms at the cost of only small increases in perplexity.

Moreover, in many cases optimizing against one detector yields a model that is also less detectable under other detectors. In particular, we find that models pre-trained against the public **RoBERTa-large** achieve an average of 0.15 reduction in AUROC when evaluated by a number of black-box commercial detectors. These results hold even at longer sequences, such as generating essays, where our fine-tuned Llama-7b-chat model achieves a RoBERTA-large AUROC of 0.26.

The results of our red-teaming effort suggest that fine-tuning language models to be less detectable is both easy and cheap, which makes it feasible for a wide slate of malicious actors, even against the best open-source and commercial detectors available. Based on these results, we advise various stakeholders (educators, policymakers etc) against reliance on the current suite of text-detectors, and to suitably account for less-detectable LLM generated text.

## 2 RELATED WORK

Machine-generated text detection methods often either train a classifier using a dataset of LM- and human-generated text (Bakhtin et al., 2019; Solaiman et al., 2019; Uchendu et al., 2020; Ippolito et al., 2020; Verma et al., 2023) or detect zero-shot by leveraging the suspected language model or a proxy (Solaiman et al., 2019; Gehrmann et al., 2019; Mitchell et al., 2023; Su et al., 2023). Prior works have called into question the robustness of these detectors, finding that detectors are susceptible to paraphrasing attacks (Sadasivan et al., 2023; Krishna et al., 2023) and can perform poorly for text written by non-native speakers Liang et al. (2023).

Further, Mitchell et al. (2023) and Mireshghallah et al. (2023) show that zero-shot detectors show significantly reduced performance when the generating model is not known. By showing that it is

straightforward to optimize against current detectors, our results complement these prior studies, while continuing to suggest that machine-generated text detectors are not robust.

Another class of works have aimed to train language models that produce subtle 'watermarks,' i.e. indications that they were generated by a machine (Kirchenbauer et al., 2023; Zhao et al., 2023; Kuditipudi et al., 2023; Yoo et al., 2023). However, the premise of watermarking relies on the fact that all strong models in the LLM ecosystem are watermarked (i.e., hosted behind APIs that enforce watermarking); a single strong LLM with freely-available weights violates this threat model. We consider a stronger threat model where an adversary is fine-tuning the model to be undetectable.

Finally, Solaiman et al. (2019) train a detector to discriminate between human samples and samples generated by a pre-trained model (GPT-2). In our case, fine-tuning that pre-trained model to maximize the 'human' probability of the detector with DPO (Rafailov et al., 2023) is very similar to performing one round of generator improvement in a generative adversarial network (GAN; Goodfellow et al. (2014)). While adversarial objectives are typically avoided for text data due to the difficulty of differentiating through the discrete sampling step, Yu et al. (2017) and Fedus et al. (2018) show that GAN language models can produce more realistic-looking samples than MLE when evaluated by humans. Their results provide some precedent for our GAN-like training procedure for our use case of generating human-looking samples.

## 3 OPTIMIZING AGAINST LANGUAGE MODEL DETECTORS

We leverage recent advantages in fine-tuning language models with reinforcement learning to directly optimize for detector confusion. First, we review key aspects of existing methods that we will leverage for detector evasion. Then we will outline our pipeline for optimizing against detectors.

**Reinforcement Learning for Language Modelling.** We consider a language model $\pi_\theta$ that is conditioned on a prompt $x$ and auto-regressively generates an output sequence $y$. Reinforcement learning has recently emerged as a powerful technique for aligning pre-trained language models to a particular objective, which is expressed through a reward function $r(x, y)$ that assigns higher rewards to more desirable responses. In practice, the most commonly-used objective includes a KL-divergence penalty between the language model and its initialization, which in combination yield

$$\max_{\pi_\theta} \mathbb{E}_{x \sim \mathcal{D}_p, y \sim \pi_\theta(y|x)} \big[ r(x, y) - \beta \mathbb{D}_{\text{KL}} \big[ \pi_\theta(y \mid x) \,||\, \pi_{\text{ref}}(y \mid x) \big] \big], \tag{1}$$

where $\mathcal{D}_p$ is some dataset of prompts, $\pi_{\text{ref}}$ is the reference model, usually the pre-trained language model before the RL fine-tuning phase, and $\beta$ is a coefficient that controls the trade-off between reward and divergence (Ouyang et al., 2022; Bai et al., 2022; Stiennon et al., 2020). The objective above seeks to align the model with the reward function, while not deviating too far from the pre-trained reference model. This KL-constraint is used in part because, without proper regularization, reward maximization alone often leads to 'reward hacking' (Skalse et al., 2022) or 'overoptimization' (Gao et al., 2022); our experiments verify that overoptimization can be an issue in our setting as well, justifying the KL-constrained formulation.

Several algorithms have been developed to optimize the objective, the most widely used one being PPO (Schulman et al., 2017). However, these algorithms are quite complex, depending on a wide set of parameters and often being unstable to train (Zheng et al., 2023).

**Direct Preference Optimization (DPO).** Rafailov et al. (2023) recently proposed the DPO algorithm with the goal of enabling simpler, stabler optimization of the above KL-constrained objective in the case where the reward function is *learned* from a dataset of preference pairs. Assume a dataset of preference pairs $\mathcal{D} = \{x^{(i)}, y_w^{(i)}, y_l^{(i)}\}_{i=1}^N$ including prompts $x$ and two generations $y_w$ and $y_l$. In this notation $y_w$ is preferred over $y_l$ (denoted $y_w \succ y_l$) and the probability is modeled as a Bradley-Terry model (Bradley & Terry, 1952), written as

$$p(y_w \succ y_l) = \sigma((r(x, y_w) - r(x, y_l))), \tag{2}$$

where $\sigma$ is the standard logistic function for some reward function $r(x, y)$. The Direct Preference Optimization algorithm (Rafailov et al., 2023) shows that the exact optimal policy $\pi^*$ for the problem in Eq. 1 can be directly optimized through the MLE objective

$$\mathcal{L}_{\text{DPO}}(\pi_\theta; \pi_{\text{ref}}) = -\mathbb{E}_{(x, y_w, y_l) \sim \mathcal{D}} \left[ \log \sigma \left( \beta \log \frac{\pi_\theta(y_w \mid x)}{\pi_{\text{ref}}(y_w \mid x)} - \beta \log \frac{\pi_\theta(y_l \mid x)}{\pi_{\text{ref}}(y_l \mid x)} \right) \right]. \tag{3}$$

| | | | | | Detector Trained Against | | |
|---|---|---|---|---|---|---|---|---|
| | | None | RoB-lg | RoB-base | Log Prob | Log Rank | DetectGPT | DetectLLM |
| | Perplexity | 8.7 | 9.0 | 8.9 | 9.0 | 9.7 | 9.5 | 9.5 |
| *Eval Detector* | RoB-lg | 0.84 | 0.63 | 0.68 | 0.86 | 0.58 | 0.87 | 0.89 |
| | RoB-base | 0.78 | 0.55 | 0.53 | 0.78 | 0.47 | 0.79 | 0.82 |
| | Log Prob | 0.69 | 0.61 | 0.59 | 0.32 | 0.55 | 0.52 | 0.58 |
| | Log Rank | 0.75 | 0.67 | 0.65 | 0.42 | 0.61 | 0.59 | 0.63 |
| | DetectGPT | 0.81 | 0.81 | 0.80 | 0.70 | 0.81 | 0.47 | 0.55 |
| | DetectLLM | 0.82 | 0.83 | 0.83 | 0.77 | 0.84 | 0.54 | 0.61 |
| | Median AUROC | 0.80 | 0.65 | 0.67 | 0.74 | 0.60 | 0.53 | 0.62 |

Table 1: **Cross-detector generalization for open-source detectors.** For each detector, we train **3 models** with different KL constraints ($\beta \in \{0.05, 0.5, 5\}$). We report perplexity (on OpenWebText) and AUROC for the strongest resulting model (i.e., lowest AUROC) **within a perplexity budget of 9.7**, or an increase of 1.

This objective allows us to optimize $\pi_\theta$ with a classification loss and very few hyper-parameter and design choices, resulting in stable training. We therefore adopt DPO as our optimization algorithm.

We note that we also attempted using detector scores directly as rewards in Eq. 1 and optimizing with general purpose reinforcement learning algorithms, but found performance to be much worse.

**Preference Data Generation and Optimization for Detector Evasion.** Our goal is to understand how easy it is to fine-tune a pre-trained language to be undetectable by current machine-generated text detectors. In order to apply DPO to this detector-evasion setting, we construct a preference dataset using samples generated by Llama-2-7B. We generate a pair of samples $y^{(i)}, \bar{y}^{(i)}$ for each prompt $x^{(i)}$ in the dataset, using temperature 1.0.[1] Preference labels are generated by comparing the detector's 'human-ness' score $s(x, y)$ for a pair of responses, and assigning the label $y^{(i)} \succ \bar{y}^{(i)}$ if $s(x, y^{(i)}) > s(x, \bar{y}^{(i)})$; otherwise we have $\bar{y}^{(i)} \succ y^{(i)}$. Generating preferences through hard thresholding was previously adopted in (Gao et al., 2022) and we also found it to perform well in our experiments, so we keep this configuration for convenience. Once we have generated the preference dataset, we fine-tune Llama-2-7B using the DPO objective in Eq. 3.

## 4 EXPERIMENTS

We conduct a wide variety of experimental evaluations in order to understand the extent to which optimizing against detectors is feasible and cost-effective. In Section 4.1, we investigate *the extent to which training against one detector provides evasion from other detectors*, using both open-source and commercial detectors available only through APIs. Section 4.3 studies how many queries to a detector are necessary to collect a dataset sufficient for evasion. We explore detector evasion in language models fine-tuned for dialogue in Section 4.4, which explores evasion using off-policy data in a case study in essay writing. Section 4.5 provides additional experiments in the setting of creative writing. See Appendices B, C.1, and D for a study of the impact of base model size on evasion, a study of the impact of sequence length on detection, and a human evaluation of model sample quality, respectively.

**Models.** Our experiments use the open-source 7B parameter Llama-2-base model pre-trained on a large, diverse text dataset (Touvron et al., 2023) in Sections 4.1-4.3 and the corresponding Llama-2-chat model, which is fine-tuned with reinforcement learning from human feedback Christiano et al. (2017); Ouyang et al. (2022) in Section 4.4. For the off-policy data experiment in Section 4.4, we sample demonstration responses from `gpt-3.5-turbo-0613`.[2] We sample the model for up to an average of approximately 120 tokens for Sections 4.1 and 4.3, approximately 120 or 240 tokens in the sequence length ablation in Appendix C.1, approximately 250 tokens for the essay generation experiment, and approximately 250-500 tokens for the creative writing case study.

**Datasets.** In Sections 4.1-4.3, we generate texts for detection that continue short 8-token prefixes of OpenWebText documents (Gokaslan & Cohen, 2019). For the experiments with chat-tuned models

---

[1]The number of prompts varies; typically it is on the order of 10k. The specific value is noted where relevant.
[2]https://platform.openai.com/docs/models/gpt-3-5.

| | | *Detector Trained Against* | | | | | |
|---|---|---|---|---|---|---|---|
| | | **None** | **RoB-lg** | **GPTZero** | **Winston** | **Originality** | **Sapling** |
| | **Perplexity** | 8.7 | 9.0 | 9.5 | 9.6 | 9.7 | 9.3 |
| *Eval Detector* | **RoB-lg** | 0.84 | 0.63 | 0.78 | 0.83 | 0.64 | 0.76 |
| | **GPTZero** | 0.57 | 0.63 | 0.57 | 0.60 | 0.62 | 0.61 |
| | **Winston** | 0.53 | 0.46 | 0.30 | 0.20 | 0.40 | 0.43 |
| | **Originality** | 0.99 | 0.69 | 0.79 | 0.82 | 0.62 | 0.79 |
| | **Sapling** | 0.75 | 0.76 | 0.77 | 0.78 | 0.75 | 0.74 |
| | **Median AUROC** | 0.75 | 0.63 | 0.77 | 0.78 | 0.62 | 0.74 |

Table 2: **Cross-detector generalization for commercial detectors.** For each detector, we again train **3 models** with different KL constraints ($\beta \in \{0.05, 0.5, 5\}$) and select the strongest as in Table 1. We report the perplexity (computed on OpenWebText) and the AUROC of each resulting model.

in Section 4.4's essay-generating case study, we use prompts from the Alpaca instruction dataset (Taori et al., 2023) for generic evasion tuning and a collection of free, human-written essay titles (along with model-generated essays) for essay-specific evasion-tuning (Vechtomov, 2023). For the creative writing case study, we use human-written prompts collected from Reddit and compiled in the Writing Prompts dataset (Fan et al., 2018). The protocol that we use to split the datasets is described in Appendix A

**Detectors.** We conduct experiments with a variety of open-source and commercial detectors. We use strong open-source detectors from prior work, including language models trained for detection such as the **RoBERTa-large** and **RoBERTa-base** models trained in Solaiman et al. (2019) as well as 'zero-shot' detectors (Mitchell et al., 2023) such as **Log Rank** and **Log Probability** thresholding (based on the intuition that model samples are assigned higher probability by the generating model, and likewise lower average per-token rank in each conditional distribution). Finally, we include zero-shot perturbation-based methods **DetectGPT** (Mitchell et al., 2023) and **DetectLLM** (Su et al., 2023), which perform detection by measuring the change in log probability or log rank of a passage after it is perturbed (i.e., slightly rephrased). For text generated by a model, slight perturbations significantly decrease the likelihood of the text under the model, whereas perturbing human-written text tends not to change log probability significantly.

For zero-shot detectors, we make the assumption that the detector knows that samples are being generated by a Llama-2 model; that is, for these detectors, we use Llama-2-base (the model we fine-tune) to compute log probabilities and ranks. This configuration represents an optimistic case for the detector. In addition to open-source detectors, we train against four popular commercial detectors, **GPTZero**, **Sapling**, **Originality.ai (model 2.0)**, and **Winston AI**.[3] All commercial detectors advertise strong performance against widely-used LLMs.

## 4.1 EVALUATING GENERALIZATION OF DETECTOR EVASION ACROSS DETECTORS

In our first experiment, we study the basic question of the feasibility of optimizing language models against language model detectors without significantly harming sample quality. We fine-tune three Llama-2 7B models on preferences computed from a variety of open source (Table 1) and commercial (Table 2) detectors; each of the three models uses a different $\beta$ parameter for DPO in the set, which corresponds to the strength of the KL regularization. We use $\beta \in \{0.05, 0.5, 5\}$.

For each set of 3 runs, we select the run that produces the **lowest** AUROC on the detector it was trained against (i.e., evades most successfully) such that the perplexity of the resulting model on OpenWebText increases by no more than 1. This selection procedure produces a single model from each detector trained against; we evaluate each model (and the original pre-trained model) against all detectors. We train each model for up to 30k preference pairs (selecting the lowest AUROC checkpoint at 10k increments). For RoBERTa-large, RoBERTa-base, log probability, and log rank, we train for a single epoch of 30k examples; for the more computationally expensive (DetectGPT, DetectLLM) or costly (commercial detector APIs) detectors, we train for up to 3 epochs of 10k preference pairs.

---

[3]https://gptzero.me/; https://sapling.ai/; https://originality.ai/; https://gowinston.ai/

| $\beta$ | Perplexity | AUROC | Model Sample |
|---|---|---|---|
| 5 | 9.5 | 0.16 | *Abstract Neuroimaging evidence suggests that the* two hemispheres may respond to stimuli somewhat differently at the behavioural level. But whether this is a general [...] |
| 0.5 | 10.0 | 0.14 | *Abstract Neuroimaging evidence suggests that the* human brain is capable of detecting patterns at different scales with distinct networks in the cortex [...] |
| 0.05 | 29.8 | 0.06 | *Abstract Neuroimaging evidence suggests that the* two 'lost' British Army battalions at Singapore fell foul of Zener Cards Very minor 90th Anniversary yesterday of the [...] |

Table 3: **An insufficient KL constraint leads to reward overoptimization and ultimately model degradation.** We compare a sample from each of the three models fine-tuned against GPTZero; for higher $\beta$ (stronger KL constraint), we see lower OpenWebText perplexity and higher AUROC (though the AUROC is still well below 0.5). While lower $\beta$ leads to even greater confusion of the detector (i.e., better reward exploitation), it leads to nonsensical generations.

We find that optimizing against both open source and commercial detectors is effective, driving the AUROC of the detector trained against to 0.63 or below (i.e., close to random) in nearly every case (the exception being Sapling, which has a relatively weak AUROC of 0.75 even before optimization). In addition, we find strong generalization between detectors when training against the OpenAI RoBERTa-large detector, DetectGPT/DetectLLM, log rank detection, or Originality. Performance is, intuitively, correlated for open-source detectors with similar mechanisms: RoB-lg and RoB-base are both fine-tuned RoBERTa models; DetectGPT and DetectLLM both measure a discrepancy that appears after perturbing the given text. Each of these pairs displays strong transfer. Similary, among the commercial detectors, we note similar performance from the RoBERTa and Originality detectors; this result is perhaps unsurprising because Originality also uses a fine-tuned RoBERTa-like model for detection.[4]

### 4.1.1 DETECTOR STRENGTH AND TRANSFER

When examining transfer of detectors, we found that the models which achieved a very low AUROC against the detector against which they were trained (log probability and Winston, with AUROCs of 0.32 and 0.20, respectively) had very poor evasion against the suite of detectors as a whole (median detector AUROCs of 0.74 and 0.78, respectively). Another group of models did not achieve such extremely low AUROCs against the detectors against which they were trained (RoB-lg, log rank, and Originality, with AUROCs of 0.63, 0.61, and 0.62, respectively). However, these models did significantly better against the suite of detectors as a whole (median detector AUROCs of 0.65, 0.60, and 0.62, respectively). The model trained against DetectGPT was a notable outlier from this trend, achieving stronger performance (median AUROC of 0.53, the lowest iof any model) under the suite of open-source detectors

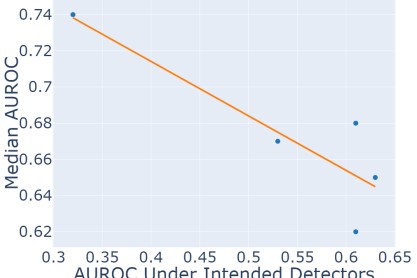

Figure 2: **Optimizing against stronger detectors yields better transfer to other detectors.** The lower the AUROC of the detector after optimizing against it (i.e. detector weakness), the higher the median AUROC of the suite of open source detectors. DetectGPT is ommitted, as an outlier.

than its performance against DetectGPT itself (AUROC reduced to 0.47) would predict. The other 5 models in the open-source suite are shown in Figure 2.

Overall, these results suggest that the more easy it is to optimize against a model, the less useful this optimization is against other detectors. **Conceptually, this implies that the following two measures of detector strength are consistent:** First, a 'strong' detector is one that is difficult to optimize against. Second, a 'strong' detector is one that, when optimized against, produces strong transfer of evasion capacity to other detectors (i.e., the features it uses are common to any other accurate detector). Thus, we find that **models fine-tuned against the strongest detectors (i.e., the ones for which decreasing the AUROC is most difficult) generalize best to other detectors.**

---

[4]https://originality.ai/blog/ai-content-detection-accuracy

| Beta | AUROC of RoBERTa Large | PPL | Fluency Win Rate | Offline Entropy |
|------|------------------------|-----|------------------|-----------------|
| 0.05 | 0.28 | 11.7 | 41.1% | 0.1523 |
| 0.5 | 0.62 | 9.0 | 52.1% | 0.1556 |
| 5 | 0.71 | 8.9 | 54.6% | 0.1564 |
| $\infty$ (base) | 0.84 | 8.7 | 50.0% | 0.1523 |

Table 4: Optimizing against RoBERTa Large with three different beta parameters. As the constraint to the original model is loosened, generated text quality and offline entropy increase then decrease while perplexity over human text monotonically decreases.

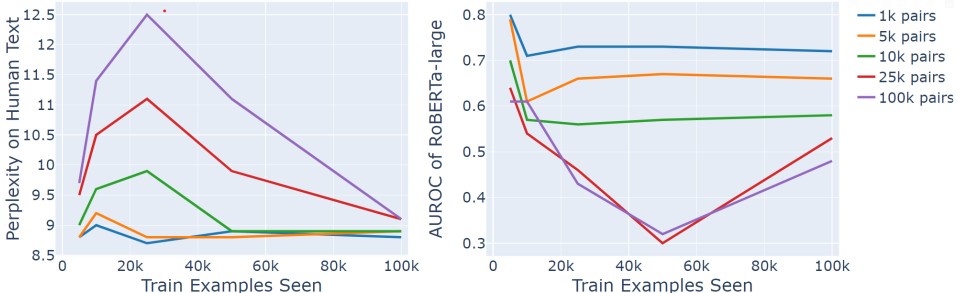

Figure 3: **Only a small number of detector queries are needed to produce a dataset large enough to confuse even a strong detector (RoBERTa-large).** Just 10k queries (to create a dataset of 5k pairs) enables meaningful optimization against the detector within a small perplexity budget; that is, gathering a training dataset of detector scores suitable for fine-tuning a language model to evade the detector is fast and economical.

## 4.2 TEXT QUALITY AFTER FINE-TUNING

In previous sections, we consider perplexity over human text as a general metric of text quality. However, in order to examine how the quality of text generated by a model is impacted by this fine tuning process, we introduce two new metrics, GPT4 win rate and offline entropy. The win rate is the percentage of time that GPT4 believes that a generation by the fine-tuned model is more "fluently and coherently written" than a generation from the base model with the same prompt. The offline entropy is the model entropy, evaluated over human text to conserve compute, and is used to evaluate the diversity of model outputs. We compute these metrics for three 7-billion parameter Llama2 models trained against the large OpenAI detector, each with a different beta parameter. The results can be found in Table 4.[5]

We note that fine tuning with our algorithm with a sufficiently strong KL constraint to the base model actually improves both text diversity and quality according to GPT-4 while still evading the detector to a significant degree. As the KL constraint becomes extremely loose, evasion is improved but text quality degrades. However, in a human evaluation in Appendix D, we find a similar-sized effect in the opposite direction, suggesting model quality according to humans may have slightly degraded.

## 4.3 IMPACT OF DETECTOR QUERY QUOTA ON EVASION

In a real-world scenario, a malicious actor is likely to be constrained by the amount of paired preference data used in training, especially if optimizing against a commercial detector. We demonstrate that strong performance of detectors can be maintained even when training data is limited. We trained five llama2-7b models against OpenAI's large RoBERTa-based detector, using $\beta = 0.5$ for all models. Each model was trained for up to 100k steps on a different training set size (1k, 5k, 10k, 25k, and 100k preference pairs). We observe several notable findings, presented in Figure 3.

First, detector evasion is possible with only a small number (<10k) queries to the detector. For the commercial detectors we study, this number of queries typically costs less than \$150 (in several

---

[5]The setup for this experiment is slightly different than the others presented in this paper. While most experiments test the post-evasion models by generating completions on a different topic distribution than the topic distribution training data was generated from, this experiment trains and tests on the *same distribution of topics*. The subsequent dataset and model size ablations are set up similarly.

cases significantly so), making detector evasion a very accessible procedure even for small budgets. Second, while AUROC of the targeted detector generally trends downward with increasing dataset size, the perplexity of the evasion-tuned model is non-monotonic. In summary, these results suggest that preventing an adversary from collecting a dataset of detector evaluations large enough to train an undetectable model may be extremely difficult or impossible.

## 4.4 GENERAL-PURPOSE ESSAY GENERATION

To analyze the feasibility of optimizing against existing detectors in a domain more closely aligned with real-world usage, we explore evasion tuning in the context of generating essays. We ask two questions: *Can we evasion-tune a chat model to generate essays that confuse a strong detector?* and a more challenging question *Can we evasion-tune a general-purpose dialogue model that still successfully confuses a strong detector in the context of essay generation, but without gathering new detector-annotated preferences or fine-tuning specifically for essay generation?* While an affirmative answer to either question is cause for concern, an affirmative answer to the second is much more serious: in this case, we do not need to evasion tune again for each new domain in which we would like to evade a detector, and further, in order to do so, we can re-use a single set of preference data generated by another model (in this case, ChatGPT).

|  | Source Model | | |
|---|---|---|---|
| **Metric** | **Base** | **Essays** | **Dialogue** |
| **AUROC** | 0.83 | 0.26 | 0.43 |
| **Perplexity** | 6.0 | 7.0 | 7.0 |

Table 5: A case study in generating difficult-to-detect essays from Llama-7b-chat. We perform detector evasion tuning on preferences generated by RoBERTa-large.

To answer these questions, we perform detector evasion on a LLama-2-chat 7B (Touvron et al., 2023), using two different datasets, one specifically essay prompts and essays generated by Llama-2-chat, and the other more general instruction-following prompts from Alpaca (Taori et al., 2023) and preference data over samples from ChatGPT, rather than the Llama-2-chat model. We fine-tune both models for 30k steps using $\beta = 0.5$; the Llama-generated preference samples are 250 tokens long; we prompt ChatGPT to write a 'single mid-length paragraph' on the given topic, discarding samples less than 100 Llama-2 tokens.

The results in Table 5 show that optimizing against RoBERTa-large is successful in both of these cases: **fine-tuning a general-purpose chat model to evade a detector using general-purpose instruction following prompts and off-policy samples nonetheless can evade a detector in the specific case of generating essays.**

## 4.5 CASE STUDY: CREATIVE WRITING IN THE CLASSROOM

Finally, we evaluate this pipeline in a setup mimicking a real classroom environment. We select the common classroom task of creative writing and the most robust commercial detector based on previous experiments, Originality. Purely creative activities have been recommended to teachers as a way to limit AI use,[6] making a simple attack on detectors in this domain particularly threatening.

We source the prompts and human completions from the Writing Prompts dataset (Fan et al., 2018), which consists of content sourced from Reddit's r/WritingPrompts subreddit.

We use out-of-distribution data generation with GPT4 to create training data that includes a wide diversity of Originality humanness scores. We trained the Zephyr 7b chat model, as its evaluations suggest it is comparatively very strong for its size (Tunstall et al., 2023). We found that the fine-tuned Zephyr chat model was able to achieve a significant reduction both in Originality's detector AUROC and in the true positive rate at a fixed low false positive rate of 1%, seen in Table 6. In practice, the true-positive rate only at a very low false-positive

| Model | AUROC | TPR (@1% FPR) |
|---|---|---|
| Base | 0.94 | 89% |
| Fine-Tuned I | 0.85 | 32% |
| Fine-Tuned II | 0.71 | 23% |

Table 6: Fine-tuning a small chat model for only ~2 hours to write creative essays while evading the Originality detector acheives a significant reduction in AUROC. More importantly, it reduces the number of AI-generated essays the detector can catch with a fixed, low false-positive rate *by a factor of three* after one round of DPO (fine-tuned I; similar to prior experiments). Fine-Tuned II generates a new preference dataset with the Fine-Tuned I model and performs DPO again.

---

[6] https://teaching-resources.delta.ncsu.edu

rate is meaningful, rather than the AUROC.
For example, a $1\%$ false-positive rate produced
enough false positives to convince Vanderbilt
University to disable their AI detector entirely,[7]. After this first round of fine-tuning, the resulting model is capable of producing quality preference pairs, eliminating the need for out-of-distribution training data. We used this fine-tuned model as a base model for a second iteration of optimization. The results can be found in Table 6.

## 5 DISCUSSION

This paper presented an empirical investigation on the ease of optimizing for language models that cannot be easily detected by current machine-generated text detectors. Our investigation is motivated by the increasingly widespread use of large language models. While language models have legitimate uses in a wide variety of applications, their use has also raised concerns that stem from both malicious intentions (e.g. cheating, disinformation) and the model's own shortcomings (e.g. generating false statements in high-stakes domains like law and medicine). Because of these concerns, a number of methods have been developed to detect machine-generated text, including commercial offerings. One company offers the "most trusted" detector, boasting a 99.6% accuracy (Winston AI, 2023), while another claims to be "the gold standard in AI detection" (GPTZero, 2023). Users have therefore put more trust in these detectors than perhaps is warranted (Thompson & Hsu, 2023).

Our experiments expose a notable weakness of these detectors: it is straightforward to fine-tune a model to evade these detectors while still maintaining high performance. The fine-tuned models produce text that is almost completely undetectable by two out of four commercial detectors. For the other two commercial detectors, the fine-tuned models have an AUROC of less than 0.5, indicating that they produce text that is judged to be statistically *more* likely to be human than the human-written corpus itself. In a hypothetical scenario such as an interaction between a detector and an online misinformation campaign, relying on such a text detector would actually hinder the verification efforts. Moreover, generating long-form text, such as essays, does not increase detectability.

We emphasize that the this training pipeline is straightforward and easy for adversaries to replicate. It uses easily accessible public models and an open-source training codebase. The entire data acquisition and training process cost a few hundred dollars. For data, the process only requires limited, black-box query access to the detector with a budget of a few thousand prompts and does not need human annotators, paraphrasing, or teacher models. For compute, we used widely available consumer hardware and only a few hours of training time. We further expect that, with more extensive data, training, and resources, a fine-tuned model may be even more evasive.

We expect that this direct kind of attack is hard to protect against. Indeed, our results showed meaningful transfer between strong detectors, and as discussed in Appendix C, obvious countermeasures each come with a nontrivial downside. Thus, in light of these results, we argue that the current generation of machine-generated text detectors is not robust to adversaries and may even favor machine-generated text over actual human-generated content. This includes both public detectors and closed black-box commercial ones. Furthermore, we argue that the problem of robust machine-generated text detection may be unsolvable in practical settings. Any new detection algorithm can be subject to the adversarial training process in this paper. New detection algorithms will be rendered ineffective by further model fine-tuning, which would then require the development of new detection algorithms. Hence, we argue against continued use of machine-generated text detectors.

---

[7]https://www.vanderbilt.edu/brightspace/2023/08/16/
guidance-on-ai-detection-and-why-were-disabling-turnitins-ai-detector/

## ETHICS STATEMENT

Evading language model detectors is a type of red-teaming exercise that we carry out in order to call attention to the serious risks of relying on any machine-generated text detection technologies. We categorically do **not** advocate for evading language model detectors for the purpose of carrying out harmful activities with LLMs. Rather, we hope that in demonstrating the ease with which the effectiveness of existing detectors can be severely degraded, we can spur a conversation about these technologies. Ultimately, we believe swift action to revise institutional norms, particularly standards in classrooms around student assessment, is warranted.

## REPRODUCIBILITY

Section 3 covers the details of the direct preference optimization algorithm, and an open-source implementation is available in the cited paper. Precise descriptions of the fine-tuning and model selection process for non-chat models are available in sections 4.1, 4.3, and C.1, while the corresponding information for chat models can be found in section 4.4. A precise description of the data generation process can be found in Appendix A.

Models, datasets, and selected experiment code will be made available at `https://github.com/charlottttee/llm-detector-evasion`. We intend to release all models presented in this paper. More specifically, we intend to release:

- All of the 7-billion-parameter Llama2 models presented in Table 1 and Table 2.
- The three variants of the 7-billion-parameter Llama2 model optimized against RoBERTa Large with different KL-constraints to the baseline that are presented in Table 4.
- The five variants of the 7-billion-parameter Llama2 model optimized against RoBERTa Large with different training dataset sizes that are presented in Figure 3.
- The two 7-billion-parameter Llama2 chat models trained on essays and out-of-distribution (ChatGPT-generated) dialogue that are presented in Table 5.
- The 13-billion-parameter Llama2 model presented in Table 7.
- The 7-billion-parameter Zephyr chat model presented in Table 6.

We additionally intend to release the datasets we annotated for this project. More specifically, we intend to release:

- A dataset of pairs of generations from the base 7-billion-parameter Llama2 model with the associated detector scores and winner assigned by each of the open-source and commercial detectors evaluated in Table 1 and Table 2.
- For each model featured in Table 1 or Table 2, a smaller dataset of its generations annotated by each detector.
- A list of longer generations from the 7-billion-parameter Llama2 model trained against RoBERTa Large, annotated by RoBERTa Large.
- The detector-annotated, ChatGPT-generated dataset used in dialogue training (see Table 5).
- The detector-annotated, Llama2-generated dataset used in essay training (see also Table 5).
- A dataset of pairs of outputs of 13- and 7-billion-parameter post-evasion Llama2 model outputs, annotated with their GPT4-generated comparison.
- The dataset of human annotations discussed in Section D.

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

## A    DATA SOURCES AND PARTITIONING

To obtain training and testing data, we load the first 110k texts from the openwebtext dataset. Of these, we randomly select 10k evaluation texts. We then save both evaluation and training prompts, which are the first 8 GPT-2 tokens of these texts. We also save the human completions of the evaluation prompts. All evaluations are done using completions to the eval prompts (including the human completion, which is truncated to approximately match the token count of the data it is being compared to in any given metric, and model-generated completions).

All training is done using model-generated completions to the training prompts. Each prompt is completed twice by the base model, and a detector is used to select the "more human" completion. These two completions then form a single preference pair in the training set. No 8-token prompt is used to create more than one preference pair. Not all train prompts are used to train the model; in most cases 30k are used. The data use experiment explores the effect of using a different amount of this data.

We follow the same procedure in the essay setting, using titles rather than prompts and asking a chat model to write an essay with said title rather than using a purely generative model to continue an 8-token prompt. To source these titles and human completions to evaluation-set titles, we use the ivypanda-essays dataset.

For the out of distribution setting, the procedure is identical, except training pairs are not generated from the model we fine tune but instead by GPT3.5 (with varied temperatures), effectively creating an off-the-shelf dataset that is model-agnostic. For this setting, we use the alpaca instruction-following dataset, and we prune training examples where either response in the preference pair is less than 250 characters or either response contains the phrase "an AI".

## B    IMPACT OF MODEL SCALE ON EVASION AND FLUENCY

We repeated the fine tuning process against the large, RoBERTa-based OpenAI detector with the 13-billion parameter Llama 2 model (using a beta parameter of 0.05). Compared to the 7-billion parameter Llama 2 model trained against the same detector with the same beta parameter, we observed a slight improvement in performance against the detector, pictured in Table 7.

Additionally, we generated one passage from each of these models for 1000 prompts and asked GPT4 to select the more "fluently and coherently written" of the two. The 13-billion parameter won 72.8% of these matchups. These results suggest that increasing the size of the base model results in similar if not better performance evading the detector while the resulting model maintains superiority in quality of text generated.

| Model | Fluency Win Rate | AUROC |
|---|---|---|
| Llama2-7b | 50.0% | 0.28 |
| Llama2-13b | 72.8% | 0.26 |

Table 7: Increasing the model size slightly improves ease of optimization against the RoBERTa-large OpenAI detector. Additionally, the fluency win rate against the fine-tuned 7-billion-parameter model improves drastically as the model is scaled up.

## C    VULNERABILITIES AND COUNTERMEASURES

Understanding how to counter this kind of attack and whether such counters are feasible is an important next step in this line of research. We leave experimental work on this front to future works, but we discuss three key vulnerabilities and potential defenses against our attack below.

### PREVENTING ACCESS TO MODEL FINE-TUNING

In the settings in which detection is most widely employed (namely education and journalism), use of models with closed weights is common. A malicious actor wanting to implement our attack would be limited to models with accessible fine-tuning, which may in turn limit the quality of undetectable text such an actor could produce. Though note that fine-tuning ChatGPT recently became publicly available, so this countermeasure may be quite weak. Further, this "defense" is not within the control of the detector supplier but rather the model supplier, so this is a relatively weak framework for defense.

|              | Source Model | |
| Seq. length  | Base | Post-evasion |
| --- | --- | --- |
| $\bar{n} = 120$ | 0.84 | 0.63 |
| $\bar{n} = 250$ | 0.92 | 0.67 |

Table 8: **Sampling longer responses from an evasion-tuned model does not improve detector AUROC for RoBERTa-large.** After evasion tuning on examples 120 tokens long, we evaluate the RoBERTa-large detector on model/human samples of 120 tokens and 240 tokens; longer samples from the base model are easier to detect, but after evasion tuning, sequence length has little affect on detector AUROC.

### GATEKEEPING DETECTORS

Our method requires running every piece of training data through a detector. While the attack is quite data-efficient, and tagging a training dataset is very cheap when using most commercial detectors, this attack would be much harder if detectors were not publicly available. A detector supplier could counter our attack by releasing the detector only to trusted users (i.e. access to educators provided through their institution). However this countermeasure would likely create inconvenient barriers to legitimate users of detectors.

### BINARY SCORING

Our method requires comparing the two "human-ness" scores of generations by a model in order to form a preference pair. In the detectors we experimented with, this value was reported to several decimal places, and ties were very rare. However, if a detector were to return a binary score ("AI" or "human"), rather than a continuous scalar score, this would make our process less data-efficient, since all pairs that were both labeled as AI or both labeled as human would have to be discarded. This also reduces the diversity of score discrepancy in the training data. However, this countermeasure has a serious drawback: a binary score significantly reduces the information provided to the user. Especially in situations that require knowing the confidence of a prediction (such as where there may be repercussions to the individual claiming to have written the text), this reduces the practicality of using the detector.

Note that each of the above countermeasures come with a significant downside - the first is not within the control of the detector supplier, the second severely reduces the commercial viability of detection businesses (especially those that are direct to consumer), and the third reduces detector utility.

### C.1 ROBUSTNESS OF EVASION TO LONGER SEQUENCE LENGTH

Prior work has shown that detector accuracy improves as the length of the generated data increases (Kirchenbauer et al., 2023). We therefore conduct a small experiment to assess whether this pattern holds true after a language model has been optimized against a given detector. The results are shown in Table 8.

We draw samples from a Llama-2 7B base model before and after it has been fine-tuned with $\beta = 0.5$ to evade RoBERTa-large for 100k preference pairs. While the pre-evasion model generates samples that are increasingly easy to detect as sequence length increases, after fine-tuning, doubling the sequence length does not lead to a drastic increase in AUROC. This result shows that detector evasion fine-tuning can generalize to longer sequences than it was trained for, potentially further increasing its general applicability.

## D    HUMAN EVALUATION

In order to reliably characterize how the quality of text produced by a model post-optimization compare to those produced by the base model, we hired human evaluators via Prolific. We produced 182 pairs of text 128 Llama2 tokens long and with the same prefix. One from each pair was generated by base Llama-2, while the other was generated by the Llama-2 model fine-tuned against Originality, the strongest commercial detector in our previous experiments.

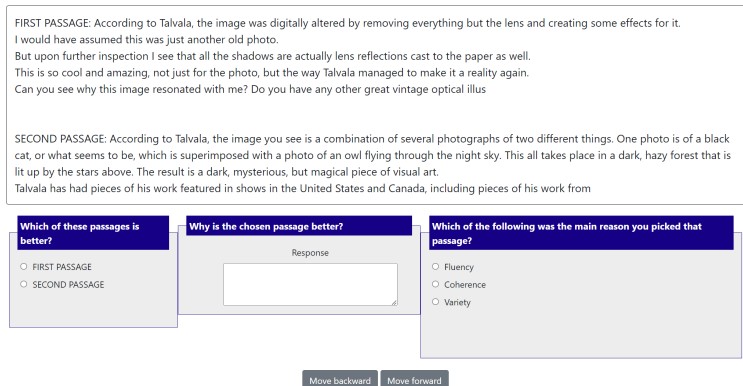

Figure 4: The human annotators were presented with two texts, one generated by the baseline model and one generated by the model fine-tuned against Originality. They were asked to select the superior text.

| Evasion Technique | AUROC of RoBERTa Large | Fluency Win Rate |
|---|---|---|
| Added Spacing | 0.71 | 13.9% |
| DIPPER Rephrasing | 0.95 | **54.3%** |
| DPO w/ Llama2-7b (ours) | **0.62** | 52.1% |

Table 9: Our method (with $\beta = 0.5$) outperforms other attacks while maintaining competitive text quality.

The two texts were presented without their sources in a randomized order to evaluators who were tasked with selecting the stronger completion. Questions about annotator reasoning as well as test questions with clear answers were supplied to ensure engagement. No significant difference in reasoning was found between those who preferred the base model and those who preferred the fine-tuned model. *Each pair of texts was annotated by two humans.*

Overall, $61\%$ of annotators selected that the base model text was superior, indicating a statistically-significant preference for the base model (with a binomial test p-value of $1.01 \times 10^{-5}$). However, in $40\%$ of cases, the annotators disagreed on the superior text (in $41\%$ of cases, both annotators preferred the baseline, and in $19\%$ of cases, both annotators preferred the post-evasion model), indicating that the difference in quality was not extreme. A qualitative look at the texts produced (see Appendix F), as well as the preference of GPT4 for the post-evasion texts (see Table 4), further suggests that the degradation in text quality is not egregious.

See Figure 4 for the interface used by labelers to annotate text quality in our human evaluation.

## E COMPARISONS WITH EXISTING ATTACKS ON DETECTORS

We have added a new experiment comparing our RL-based attack with two existing methods in the literature, including a method based on paraphrasing (Sadasivan et al., 2023) and perturbing spacing around punctuation (Cai & Cui, 2023). For each attack, we evaluate the resulting detector AUROC and the percentage of time that GPT4 ranks the adversarial text as more coherent and fluent than the original text (GPT4 Win Rate). These results can be found in Table 9. We find that our approach can achieve a unique combination of post-evasion fluency and evasion capacity as compared with existing baselines. Additionally, when compared with a paraphrasing attack, our method does not require any compute power beyond that needed to generate the sample itself - no need to run the sample through a large paraphrasing model (which in this case is larger than the original generative model producing the sample).

## F POST-OPTIMIZATION SAMPLES

Tables 10 and 11 display ten pairs of samples that are meant to provide a qualitative overview of the difference between pre- and post-optimization generations. The table compares the base 7-billion-

parameter Llama2 model to the same model optimized against the RoBERTa Large detector (the model is displayed in Table 1).

| Base Sample | Optimized Sample |
| --- | --- |
| San Salvador, Oct 29 (Prenh Channel) - According to the latest data on the economic outlook of El Salvador, inflation is expected to reach 4.3% by the end of the year, below the target that was initially expected. [newline] The report also highlighted that it expected El Salvador to record a 1.2% increase in GDP in the last three months of the year. [newline] The current inflation of El Salvador at the end of the month is recorded at 3.1%. [newline] However, it highlights that, after two years of | San Salvador, Oct 29 (Prenhdeep Kaushal) [newline] Pakistani cricket team's chief selector Inzamam-ul-Haq said on Tuesday that legendary leg-spinner Shane Warne advised him to remove bowlers' fielding catcher off the field on full extra cover point to avoid fielding errors behind the bowler's legs. [newline] Warne, who led Australia to the 2003 Cricket World Cup Final in South Africa, came up with the suggestion after Pakistan bung |
| The American team, led by Jay Penske, had an impressive season with Ferrari. [newline] The Ferrari team in 2017 was the most difficult for me. In 2010 I went into F1 knowing that I was the best driver in the world. But in 2017, I was not. [newline] The pressure was very strong. [newline] If I had stayed, things could have gone wrong. This is why I preferred to leave. In Formula 1, Ferrari has been my team since I was 10 years old | The American team, led by Jay Penske, came up one win short of the Rolex Trophy. They lost today in the final of the Rolex Match Race in Marseille. In 2000, when the event took place here for the first time, the winning team was skippered by a certain Mr Bundock.... So a home victory was very much on the cards here today.[newline] However, America had the better start winning the first two races of the best of five final. The tide turned rapidly in the third race and it was France who |
| BYU reports an exceptionally low number of students seeking alcohol and illicit drugs on campus: 0.47 percent of students reported illicit drug use and 12.7 percent reported binge drinking in 2008. BYU's strict honor code and strict rules against alcohol and drugs prohibits student from using them. However, students have reported to The Daily Universe and The Observer that these rules are frequently violated and it is easy to buy drugs and alcohol on BYU's campus as well as | BYU reports an exceptionally low number of students seeking alcohol and illicit drug counseling: 0.47% [newline] Compared to the national average, which is about 1,245.9 times higher [newline] The percentage of BYU's full-time students enrolled in courses while living on-campus is 100%.[newline] That is about 951.7 times higher than the national average of 0.10% [newline] The percentage of students enrolled exclusively in distance courses is 0.13% |
| GAZA CITY (Ma'an) – Israeli navy opened fire, on Monday, at Palestinian fishermen's boats off the coast of Rafah City in the southern besieged Gaza Strip and detained four fishermen, a witness told Ma'an. [newline] A witness told Ma'an that Israeli gunboats opened fire towards Palestinian fishermen off the coast of Rafah town. [newline] Israeli naval forces detained four Palestinian fishermen and took them to an unknown location when they were detained | GAZA CITY (Ma'an) – Israeli navy opened fire, on Monday, at Palestinian fishing boats off the northern Gaza Strip coastline, no injuries were reported. [newline] Witnesses told Ma'an the Zodiac boats Israeli navy fired upon was located ten nautical miles off the Karim Abu Sakha beach, which is near the border with Israel, east of Jabaliya in the northern Gaza Strip. They added Israeli navy fired warning shots at boats when they approached seven na |
| Bananas have been part of our diet for thousands of years. They are one of the most common tropical fruits, are nutritious, versatile, and available year-round at our local stores. But the problem is that the majority of them aren't actually bananas but rather an artificial crop: GMOs (genetically modified organisms). [newline] In this article I'll cover what you must know about the banana. First, I'll discuss the history of bananas from their native land to current industry practices. Then I'll discuss the | Bananas have been part of our diet for thousands of years. People living on the islands of Polynesia and Melanesia began cultivating bananas some 10,000 years ago. They also bred them into dozens of distinct varieties. In India, the fruit was introduced 2,500 years ago. Bananas are the most important fruit crop in Asia, and among the most important crops in Africa, Hawaii, and the Caribbean. The first bananas were green, but the yellow variety dominated around the |
| [ reblog on Tumblr ] [newline] [newline] # 6. Annotations [newline] [newline] The annotated versions of this project will be posted here. They'll be in the form of HTML files, just like this one, except the contents of the tags will be replaced by comments. [newline] [newline] These comments are the way to tell a story of your project to the world. This is where you can explain what led you to that project, what you did to work on it, what you learned along the way, and so on. [newline] [newline] This format works well for other types of | [ reblog on Tumblr ] [newline] [newline] # 6. An Autumn Stroll (with the Nose of Sherlock Holmes)[**](#contents) [newline] 1. What Did He Smell?[**](#) [newline] The autumn air was mild, [newline] With a hint of fog as well, [newline] And a smell of something sweet, [newline] Like candy made to eat... [newline] But this odor did not please [newline] Each small boy present there; [newline] It wasn't quite a nuisance [newline] Or foul as sewage |

Table 10: A random survey of pre- and post-optimization samples from a Llama-7b model trained against RoBERTa Large. There does not appear to be a significant qualitative difference between the generated texts. Continued in Table 11

| Base Sample | Optimized Sample |
|---|---|
| India will deploy the recently bought Rafale jets for operations in Pakistan and Afghanistan to strike terror camps in Balochistan, interior minister Chaudhry Nisar said here on Sunday, two days after India sent the much-awaited aircraft across. India has been maintaining that the aircraft was being bought for "self-defence" and would be deployed at bases in western coastal airfields. "We bought Rafale jets from France for operations in Pakistan (Punjab and Balochistan) and Afghanistan," said the interior minister. [newline] | India will deploy the recently bought Rafale jets for operations in Pakistan and not for patrolling its airspace alone. The deployment of the jets is being done for operations in Pakistan, he said. India will be celebrating its 74th Independence day on August 15. A large military parade is scheduled to mark the occasion, as well as a flypast by the warplanes, which are India's first from a major acquisition since 2007. Besides the Rafales that made their debut Saturday, the parade will |
| To view this video please enable JavaScript, and consider upgrading to a web browser that supports HTML5 video [newline] Posted by: Jake on: September 7, 2017 [newline] In: Entertainment — Technology [newline] Since smart phones are becoming an ubiquitous part of our life, mobile application development is becoming increasingly relevant, especially for businesses. These mobile apps provide easy, instant access to customers and consumers around the world. [newline] Business owners can easily develop, deploy and modify mobile apps to reach customers with their offerings; whether it be | To view this video please enable JavaScript, and consider upgrading to a web browser that supports HTML5 video . [newline] We love our pets so much it is hard to believe that more often than not they are often kept more like a child than an equal member of the family. [newline] However to treat them better we all know our pets and so when they start showing signs that they are not comfortable, we do all we can to help them. But there is little we can do about when they start displaying that their arthritis is starting to take its toll. [newline] The |
| Sales are up, and we haven't even launched, and we've already gotten some favorable feedback. [newline] I'm sure with a little more tweaking we can make them even better, but I think we're off to a great start. [newline] I'll try to post more pictures as our prototypes continue to improve. Thanks to everyone who has already purchased! [newline] Bamboo Garden Prototypes - Bamboo Shades! [newline] We just received our prototyped Bamboo Garden shades! We are excited to introduce | Sales are up, and we haven't started! [newline] It's official: the 2010 Fasig-Tipton Midlantic Fall Yearling Sale was the best ever. The 325 yearlings sold averaged $54,330 for a sale gross of $15,852,550. Too good to be true?? [newline] Tuesday the 13th brought 157 yearlings to the auction ring followed by another 168 on Wednesday the 1 |
| Even though PLAYERUNKNOWN'S BATTLEGROUNDS has seen its player numbers decline over the last few weeks, the popular title is far from dead. According to PUBG, the game continues to generate a revenue of over $1 million a day. And when you consider that there hasn't been a major PUBG update since December, it's clear that players are still spending a lot on microtransactions. To put this revenue into context, at the height of its popularity during 2018 PUB | Even though PLAYERUNKNOWN'S BATTLEGROUND is still to be released, GameSpot has found new footage from the game, including in-game footage and interviews with the makers. After months of silence regarding the game, PUBG has just given us a reason to be patient until release. [newline] PUBG is one of the more hotly anticipated games this year as it blends three great things in one single game; First person shooter, battle royale and a sandbox game. After the massive success Garry' |

Table 11: Continuation of Table 10.

