# OpenReview forum: "Language Model Detectors Are Easily Optimized Against"
_ICLR.cc/2024/Conference — ICLR 2024 poster_

### Official Review · Reviewer_iNUy · 2023-10-15

**Soundness:** 3 good
**Presentation:** 2 fair
**Contribution:** 3 good
**Rating:** 6
**Confidence:** 4

**Summary:**

This work mainly studies why direct preference optimization (DPO) can be used to train a generator to evade detection. Following DPO, two samples are generated, and the preference is determined by the humanness score outputed by a given detector.

In experiments, a range of detectors are optimized against, including classifiers and metric-based detectors such as DetectGPT. Empirically, AUROC metrics is reduced to below 0.5 against several strong public and commercial detectors. Also interestingly, attack against one detector could generalize to other detectors.

**Strengths:**

The attacks for detectors is a very relevant research question especially in the era of LLMs.

Within the scope of detection, using techniques from RL (DPO) is quite novel.

The attack result is quite strong, and quite concerning. Since it does not require white-box access to the detector.

**Weaknesses:**

From a ML perspective, this paper does not propose a completely novel algorithm. Therefore my rating will be higher if this is a NLP conference.

From an adversarial attack perspective, the result is not very surprising.

I think the author should not only report PPL, but also the diversity of the generated texts.

How to defense against such attack is not explored.

**Questions:**

You should put comma or period after the equations. They are also part of a sentence.

A lot of paragraphs are very long. Please break up at some points.

---

> ### Author Response · Authors · 2023-11-20
> **Addressing Algorithmic Novelty, Text Diversity, and Countermeasures**
>
> We really appreciate your review and your candid assessment of our work! To address your comments:
>
> **Algorithmic novelty.** While we agree that the purpose of our work is not to contribute a novel algorithm, we note that "societal considerations including fairness, safety, privacy" is listed as a topic of interest in the ICLR 2024 call for papers, and we feel our work fits well into this category. Given the widespread use of large language models and the numerous companies charging teachers and other machine learning laypeople money for detection services, we feel our work shares an important and timely message about the safety and reliability of widely used detection services, and that this message will be of strong interest to the ICLR community.
>
> **Text diversity.** To investigate the effects of our fine-tuning algorithm on the quality and diversity of text produced, we measure:
> 1. Entropy: computed over human text; added to measure diversity of model outputs.
> GPT4-rated Fluency and Coherence Win Rate: the percentage of time GPT4 rates the completion by the fine-tuned model as more fluent and coherent than the completion from the base model; added to measure quality of model outputs.
> 2. We computed these metrics on three models trained against OpenAI’s large RoBERTa-based detector, each with a different value of beta (KL constraint to original model).
>
> | Beta                  | Entropy | Win Rate Against Base Model | PPL  | AUROC of RoBERTa Large |
> |-----------------------|---------|-----------------------------|------|------------------------|
> | Infinite (no training)| 0.1523  | 50.0%                       | 8.7  | 0.84                   |
> | 5                     | 0.1556  | 54.6%                       | 8.9  | 0.71                   |
> | 0.5                   | 0.1564  | 52.1%                       | 9.0  | 0.62                   |
> | 0.05                  | 0.1508  | 41.1%                       | 11.7 | 0.28                   |
>
> We note that ***fine tuning with our algorithm with a sufficiently strong KL constraint to the base model actually improves both text diversity and quality while still evading the detector*** to a significant degree. As the KL constraint becomes extremely loose, evasion is improved at the expense of text diversity and quality.
>
> **Countermeasures.** We certainly agree that understanding how to counter this kind of attack and whether such counters are feasible is an important next step in this line of research. We leave experimental work on this front to future works to preserve our paper’s focus on detector fragility and this method of attack.
>
> However, we will revise the paper to include a discussion of the following three potential methods of defense and submit a revision before the end of the rebuttal period:
> 1. **Preventing access to model fine-tuning.** In the settings in which detection is most widely employed (namely education and journalism), use of models with closed weights is common. A malicious actor wanting to implement our attack would be limited to models with accessible fine-tuning, which may in turn limit the quality of undetectable text such an actor could produce. Though note that fine-tuning ChatGPT recently became publicly available, so this countermeasure may be quite weak. Further, this “defense” is not within the control of the detector supplier but rather the model supplier, so this is a relatively weak framework for defense.
> 2. Our method requires running every piece of training data through a detector. While the attack is quite data-efficient, and tagging a training dataset is very cheap when using most commercial detectors, this attack would be much harder if detectors were not publicly available. A detector supplier could counter our attack by **releasing the detector only to trusted users** (i.e. access to educators provided through their institution). However this countermeasure would likely create inconvenient barriers to legitimate users of detectors.
> Our method requires comparing the two “human-ness” scores of generations by a model in order to form a preference pair. In the detectors we experimented with, this value was reported to several decimal places, and ties were very rare. However, if a detector were to **return a binary score (“AI” or “human”) rather than a scalar score**, this would make our process less data-efficient, since all pairs that were both labeled as AI or both labeled as human would have to be discarded. This also reduces the diversity of score discrepancy in the training data. However, this countermeasure has a serious drawback: a binary score significantly reduces the information provided to the user. Especially in situations that require knowing the confidence of a prediction, this reduces the practicality of using the detector.
>
> **Punctuation and paragraphs.** Thank you for your suggestions regarding the paper presentation. We will correct these errors in a revised submission before the end of the rebuttal period!

---

> > ### Comment · Reviewer_iNUy · 2023-11-20
> > **thanks**
> >
> > thanks for the response, it's helpful.

---

### Official Review · Reviewer_oZ9u · 2023-11-01

**Soundness:** 3 good
**Presentation:** 3 good
**Contribution:** 3 good
**Rating:** 6
**Confidence:** 3

**Summary:**

The paper delves into the rising interest in identifying text generated by large language models (LLMs). Although detection systems have been implemented in various sectors, notably education, their vulnerability has been a significant concern. The authors present a data-efficient method that fine-tunes LLMs to deceive these detectors by employing the latest advancements in reinforcement learning for language models. They use the 'human-ness' score of several detectors as a reward function and set a constraint to ensure the modified model remains close to the original. Through this method, the effectiveness of the OpenAI RoBERTa-Large detector is notably reduced. The findings suggest that this enhanced 'detector evasion' can generalize to other detectors not part of the initial training. Consequently, the authors caution against depending on detectors for LLM-generated text.

**Strengths:**

- The paper is well-written and clearly presented;
- The paper tackles a critical and timely topic concerning the detection of LLM-generated text and proposed a novel data-efficient RL-based attack to deceive existing detector;
- The paper shows empirical evidence of the fragility of current LLM-based detectors, offering actionable insights for future research and development of LLM detectors, the experiments are based on three runs which show the robustness of the proposed methods, abolition regarding the sample efficiency has also been provides to show the design choices;

**Weaknesses:**

- The scalability of the proposed methods could be good to include to show whether the evasion of detector only happens when the model size of small like 7B in the most of the experiments;
- Besides the perplexity, it will be good to include some evaluations on popular benchmark to assess the post-evasion model performance (whether the improved evasion is under the sacrifice of the general performance of the attack model);

**Questions:**

- Could the authors list the training preference data and evaluation data in detail to understand whether there is a generalization due to the data in the experiments;
- Could author offer more explanation towards the generalization of cross-detectors and the mixture of the effectiveness from different sources in Table 1 and 2, as well as how the attack performance correlates with the generalization;

---

> ### Author Response · Authors · 2023-11-20
> **Addressing Scalability, Non-Perplexity Evaluations, and Training/Evaluation Data**
>
> Thank you so much for your comprehensive comments! We hope that the following will address your concerns.
>
> **Scalability: We find that increasing the model size increases evasion capability and text fluency of the fine-tuned model.**
>
> We repeated the fine tuning process against the large OpenAI detector with the 13-billion parameter Llama 2 model (using a beta parameter of 0.05). Compared to the 7-billion parameter  Llama 2 model trained against the same detector with the same beta parameter, we observed a slight improvement in performance against the detector:
>
> | Model       | AUROC of RoBERTa Large |
> |-------------|------------------------|
> | Llama2-7b   | 0.28                   |
> | Llama2-13b  | 0.26                   |
>
> Additionally, we generated one passage from each of these models for 1000 prompts and asked GPT4 to select the more “fluently and coherently written” of the two. **The 13-billion parameter won 72.8% of these matchups.** These results suggest that increasing the size of the base model results in similar if not better performance evading the detector while the resulting model maintains superiority in quality of text generated.
>
> **Non-Perplexity Evaluations: We evaluated how the fine-tuning process affects generation quality by using GPT4 fluency/coherence win rates against the base model. We found that with reasonable constraint to the starting model, fine tuning not only doesn’t cause a significant drop in generation quality, but actually improves it.**
>
> In order to examine how the quality of text generated by a model is impacted by this fine tuning process, we introduce a new metric, GPT4 win rate. This win rate is the percentage of time that GPT4 believes that a generation by the fine-tuned model is more “fluently and coherently written” than a generation from the base model with the same prompt. We compute this metric for three 7-billion parameter Llama2 models trained against the large OpenAI detector, each with a different beta parameter:
>
> | Beta | Win Rate Against Base Model | PPL  | Entropy* | AUROC of RoBERTa Large |
> |------|-----------------------------|------|----------|------------------------|
> | 0.05 | 41.1%                       | 11.7 | 0.1523   | 0.28                   |
> | 0.5  | 52.1%                       | 9.0  | 0.1556   | 0.62                   |
> | 5    | 54.6%                       | 8.9  | 0.1564   | 0.71                   |
>
> \* Entropy calculations were not included in the original paper, but have been computed in response to another review.
>
> The win rates follow a trend similar to perplexity: in order to achieve stronger detector evasion, we must reduce our KL constraint to the original model (beta parameter), which leads to poorer generation quality - both a higher perplexity and a lower GPT4 win rate.
>
> We additionally note that in two of the above cases, this kind of fine tuning actually increases the GPT4 rated fluency and coherence of the model, as evidenced by win rates over 50%. This suggests that, at least to some extent, the human-ness score of the detector correlates to text quality.
>
> **Training and Evaluation Data.** We load the first 110k texts from the openwebtext dataset. Of these, we randomly select 10k eval texts. We then save both eval and training prompts, which are the first 8 GPT-2 tokens of these texts. We also save the human completions of the evaluation prompts. All training is done using model-generated completions to the training prompts, while evaluations are done using completions to the eval prompts (including the human completion, which is truncated to approximately match the token count of the data it is being compared to in any given metric). Not all train prompts are used to train the model; in most cases 30k are used. The data use experiment explores the effect of using a different amount of this data.
>
> We follow the same procedure in the essay setting, saving titles rather than prompts and using the [ivypanda-essays dataset](https://huggingface.co/datasets/qwedsacf/ivypanda-essays).
>
> For the out of distribution setting, the procedure is identical, except training pairs are not generated from the model we fine tune but instead by GPT3.5 (with varied temperature), effectively creating an off-the-shelf dataset that is model-agnostic. For this setting, we use the alpaca instruction-following dataset, and we prune training examples where either response in the preference pair is less than 250 characters or the response contains the phrase “an AI”.

---

> > ### Author Response · Authors · 2023-11-20
> > **Addressing Cross-Detector Generalization**
> >
> > **Detector Generalization: We find that models fine-tuned against the strongest detectors (i.e., the ones for which decreasing the AUROC is most difficult) generalize best to other detectors.**
> >
> > When examining transfer of commercial detectors (Table 2), we found that the models which achieved a very low AUROC against the detector against which they were trained (GPTZero and Winston, with AUROCs of 0.16 and 0.21, respectively) had the poorest evasion against the suite of detectors as a whole (median detector AUROCs of 0.67 and 0.72, respectively). The other group of models did not achieve such extremely low AUROCs against the detectors against which they were trained (RoB-lg, Winston, and Sapling, with AUROCs of 0.62, 0.62, and 0.60, respectively). However, these models did significantly better against the suite of detectors as a whole (median detector AUROCs of 0.62, 0.62, and 0.60, respectively).
> >
> > A similar trend emerged when examining the open source suite (Table 1). The results were not so clearly clustered as the above, so we plotted them to illustrate this pattern. The model trained against DetectGPT was a notable outlier in this pattern, achieving stronger performance (median AUROC of 0.57, lower than any other model in Table 1) under the suite of detectors than its performance against DetectGPT itself (AUROC reduced to 0.48) would predict. The other 5 models in this suite are pictured [here](https://imgur.com/a/4CkYKns).
> >
> > Overall, these results suggest that the more easy it is to optimize against a model, the less useful this optimization is against other detectors. Conceptually, this implies that the following two measures of detector strength are consistent:
> > 1. A “strong” detector is one that is difficult to optimize against.
> > 2. A “strong” detector is one that, when optimized against, produces significant transfer of evasion capacity to other detectors (i.e. the features it encodes are essential to accurate detection in any method).

---

### Official Review · Reviewer_bGzn · 2023-11-03

**Soundness:** 2 fair
**Presentation:** 2 fair
**Contribution:** 2 fair
**Rating:** 6
**Confidence:** 4

**Summary:**

The paper explores the feasibility of optimizing language models to evade language model detectors. The authors propose a data-efficient attack using reinforcement learning to fine-tune language models and confuse existing detectors. They demonstrate the effectiveness of this approach by reducing the AUROC of the OpenAI RoBERTa-Large detector from 0.84 to 0.62 in a 7B parameter Llama-2 model. The results show that it is relatively easy and cheap to train language models to be less detectable, and the evasion generalizes to other detectors not used during training.

**Strengths:**

- The paper introduces a new method for optimizing language models to evade detectors using reinforcement learning. The use of direct preference optimization (DPO) and the KL-divergence constraint provides a simple and stable training procedure.

- The authors conduct a comprehensive set of experiments to evaluate the effectiveness of the proposed approach. They consider both open-source and commercial detectors, and demonstrate the generalization of evasion across detectors.

- The results of the study have important implications for the reliability of machine-generated text detectors. The findings suggest that current detectors are not robust and can be easily evaded, which raises concerns about the widespread use of language models.

**Weaknesses:**

- The paper focuses primarily on empirical evaluations and does not provide a theoretical analysis of the proposed approach. A deeper understanding of the underlying principles and limitations of the method would enhance the contribution of the paper.

- The paper does not extensively discuss potential countermeasures that could be employed to improve the robustness of language model detectors. It would be valuable to explore possible strategies for detecting and mitigating evasion attacks.

- The paper does not compare the proposed approach with existing evasion techniques. It would be beneficial to evaluate the performance of the proposed method against other state-of-the-art methods for evading language model detectors.

**Questions:**

None

---

> ### Author Response · Authors · 2023-11-20
> **Addressing Potential Countermeasures and Existing Attacks on Detectors**
>
> Thank you so much for your feedback! We hope that the following analysis and experiments can address some of your concerns.
>
> **Comparing with existing attacks on detectors.** We have added a new experiment comparing our RL-based attack with two existing methods in the literature, including a method based on paraphrasing [[1]](https://arxiv.org/pdf/2303.13408.pdf) and perturbing spacing around punctuation [[2]](https://arxiv.org/pdf/2307.02599.pdf). For each attack, we evaluate the resulting detector AUROC and the percentage of time that GPT4 ranks the adversarial text as more coherent and fluent than the original text (GPT4 Win Rate).
>
> | Evasion Technique           | GPT4 Fluency Win Rate | RoBERTa detector AUROC |
> |-----------------------------|-----------------------|------------------------|
> | DPO w/ Llama2-7b, beta = 0.5| 52.1%                 | 0.62                   |
> | Added Spacing               | 13.9%                 | 0.71                   |
> | DIPPER Rephrasing*          | 54.3%                 | 0.95                   |
>
> \* While this attack performs well against some detectors, it does not generalize as well to others, including the RoBERTa detector used in most of our experiments.
>
> We find that our approach can achieve a unique combination of post-evasion fluency and evasion capacity as compared with existing baselines. Additionally, when compared with a paraphrasing attack, our method does not require any compute power beyond that needed to generate the sample itself - no need to run the sample through a large paraphrasing model (which in this case is larger than the original generative model producing the sample!).
>
> **Countermeasures.** We certainly agree that understanding how to counter this kind of attack and whether such counters are feasible is an important next step in this line of research. We leave experimental work on this front to future works to preserve our paper’s focus on detector fragility and this method of attack.
>
> However, we will revise the paper to include a discussion of the following three potential methods of defense and submit a revision before the end of the rebuttal period:
> 1. **Preventing access to model fine-tuning.** In the settings in which detection is most widely employed (namely education and journalism), use of models with closed weights is common. A malicious actor wanting to implement our attack would be limited to models with accessible fine-tuning, which may in turn limit the quality of undetectable text such an actor could produce. Though note that fine-tuning ChatGPT recently became publicly available, so this countermeasure may be quite weak. Further, this “defense” is not within the control of the detector supplier but rather the model supplier, so this is a relatively weak framework for defense.
> 2. Our method requires running every piece of training data through a detector. While the attack is quite data-efficient, and tagging a training dataset is very cheap when using most commercial detectors, this attack would be much harder if detectors were not publicly available. A detector supplier could counter our attack by **releasing the detector only to trusted users** (i.e. access to educators provided through their institution). However this countermeasure would likely create inconvenient barriers to legitimate users of detectors.
> Our method requires comparing the two “human-ness” scores of generations by a model in order to form a preference pair. In the detectors we experimented with, this value was reported to several decimal places, and ties were very rare. However, if a detector were to **return a binary score (“AI” or “human”) rather than a scalar score**, this would make our process less data-efficient, since all pairs that were both labeled as AI or both labeled as human would have to be discarded. This also reduces the diversity of score discrepancy in the training data. However, this countermeasure has a serious drawback: a binary score significantly reduces the information provided to the user. Especially in situations that require knowing the confidence of a prediction, this reduces the practicality of using the detector.

---

### Author Response · Authors · 2023-11-23
**Revised Copy Uploaded**

Hello! We are writing to note that a PDF of the revised paper with grammatical revisions and added experiments has been uploaded. All new additions were mentioned in at least one comment in reply to a review!

---

### Meta-Review · Area_Chair_7nQi · 2023-12-13

**Metareview:**

This paper presents experiments using DPO to fine-tune an LLM to evade machine-generated text detectors. Results show that the proposed method is effective -- accuracy of detection can be brought near random with only moderate increases in perplexity -- and is transferrable -- in some settings, fine-tuning to evade one detect allows the model to also evade other detectors. Reviews are consistently in favor of marginal acceptance. The main strength brought up by nearly all reviewers: the results of this paper are worth knowing about because they decrease trust in existing detectors and may prevent over-confidence in detection accuracy. The main weaknesses discussed are related to evaluation: reviewers asked to see more evaluations of the fine-tuned LM's performance beyond perplexity on held-out human generated text. In rebuttal, automated evaluations of fluency using GPT-4 were added, but no downstream task evaluations are human evaluations.

**Justification For Why Not Higher Score:**

More thorough evaluation of performance of the resulting models (e.g. more goal-oriented task evaluations or manual human evaluations) is still missing.

**Justification For Why Not Lower Score:**

The results of this paper are worth knowing about: machine-generated text detection is relatively easy to evade.

---

### Decision · Program_Chairs · 2024-01-16

Accept (poster)